# Comparative Analysis of Plant Growth-Promoting Rhizobacteria’s Effects on Alfalfa Growth at the Seedling and Flowering Stages under Salt Stress

**DOI:** 10.3390/microorganisms12030616

**Published:** 2024-03-19

**Authors:** Xixi Ma, Cuihua Huang, Jun Zhang, Jing Pan, Qi Guo, Hui Yang, Xian Xue

**Affiliations:** 1Drylands Salinization Research Station, Northwest Institute of Eco-Environment and Resources, Chinese Academy of Sciences, Lanzhou 730000, China; maxx@lzb.ac.cn (X.M.); huangcuihua@lzb.ac.cn (C.H.); panjing@lzb.ac.cn (J.P.); 2University of Chinese Academy of Sciences, Beijing 100049, China; 3Institute of Biology, Gansu Academy of Sciences, Lanzhou 730000, China; zhangjun-204@163.com (J.Z.); guoqi9207@163.com (Q.G.); yanghui43@163.com (H.Y.)

**Keywords:** PGPR, saline–alkali soil, phosphate solubilization, indole-3-acetic acid (IAA), 1-aminocyclopropane-1-carboxylic (ACC) deaminase, salt tolerance, flowering stage

## Abstract

Alfalfa (*Medicago sativa* L.), a forage legume known for its moderate salt–alkali tolerance, offers notable economic and ecological benefits and aids in soil amelioration when cultivated in saline–alkaline soils. Nonetheless, the limited stress resistance of alfalfa could curtail its productivity. This study investigated the salt tolerance and growth-promoting characteristics (in vitro) of four strains of plant growth-promoting rhizobacteria (PGPR) that were pre-selected, as well as their effects on alfalfa at different growth stages (a pot experiment). The results showed that the selected strains belonged to the genera *Priestia* (HL3), *Bacillus* (HL6 and HG12), and *Paenibacillus* (HG24). All four strains exhibited the ability to solubilize phosphate and produce indole-3-acetic acid (IAA) and 1-aminocyclopropane-1-carboxylate (ACC) deaminase. Among them, except for strain HG24, the other strains could tolerate 9% NaCl stress. Treatment with 100 mM NaCl consistently decreased the IAA production levels of the selected strains, but inconsistent changes (either enhanced or reduced) in terms of phosphate solubilization, ACC deaminase, and exopolysaccharides (EPS) production were observed among the strains. During the various growth stages of alfalfa, PGPR exhibited different growth-promoting effects: at the seedling stage, they enhanced salt tolerance through the induction of physiological changes; at the flowering stage, they promoted growth through nutrient acquisition. The current findings suggest that strains HL3, HL6, and HG12 are effective microbial inoculants for alleviating salt stress in alfalfa plants in arid and semi-arid regions. This study not only reveals the potential of indigenous salt-tolerant PGPR in enhancing the salt tolerance of alfalfa but also provides new insights into the mechanisms of action of PGPR.

## 1. Introduction

Minqin County (Gansu Province, China), lying downstream of the Shiyang River Basin, is an essential production base for grains and cash crops, but it is also a typical area of ecological degradation in the arid regions of Northwest China [1]. Over the past few decades, the increase in soil salinization and groundwater mineralization has compelled the people of Minqin County to reduce the cultivated area of salt-sensitive crops such as wheat and cotton. Simultaneously, there has been an increase in the cultivation of salt-tolerant forage crops like alfalfa [2]. Studies have indicated that cultivating forage yields the highest profits, exceeding those derived from traditional grain crops and the combination of grain and cash crops by 56.41% and 26.42%, respectively [3,4]. Considering the growing demand for high-quality forage, utilizing saline–alkali lands for forage production is the most sustainable and economically viable approach.

Alfalfa is a moderately salt-tolerant forage crop. However, its germination rate and growth speed decrease at 50 mM NaCl [5], and its yield decreases by 7.3% with soil salinity of 2 dS·m^−1^ [6]. Therefore, enhancing the salt tolerance of alfalfa is a critical step toward achieving high yields in saline–alkali lands. A review study has shown that inoculating seeds, plant surfaces, and soil with salt-tolerant PGPR (plant growth-promoting rhizobacteria) can enhance the crop yield under stress conditions [7]. This approach offers a promising alternative for improving crop resilience to salinity, potentially enabling more sustainable agricultural practices in saline-affected areas.

The growth of PGPR strains is inhibited by salt stress [3,4,8,9], but their growth-promoting characteristics may show a trend of initial enhancement followed by a decline [4] or continuous enhancement [3,9] as salinity increases. Studies have focused on the growth-promoting effects of PGPR on specific growth stages of alfalfa, such as the seedling or flowering stage; however, there has been a notable absence of studies examining how alfalfa’s growth and physiological traits vary across different stages after inoculation with PGPR [6,10,11,12,13,14]. A comparative analysis of the various growth stages facilitated by PGPR is vital for understanding plant growth and development and devising strategies to adapt to external environmental changes. This analysis will also guide the scientific management of fertilization across different growth stages. Based on the considerations above, the objectives of this study are to (1) fully explore the diverse plant growth-promoting traits of the selected PGPR strains; (2) compare the changes in the plant growth-promoting traits of PGPR under normal and salt stress; and (3) compare the differences in the growth and physiological characteristics of alfalfa inoculated with PGPR strains at different growth stages.

## 2. Materials and Methods

### 2.1. Isolation and Identification of Selected Strains

We collected the rhizosphere samples of three typical halophytes (*Kalidium foliatum*, *Lycium ruthenicum*, and *Tamarix ramosissima*). Then, 1 g of rhizosphere soil was weighed and placed in a test tube containing 9 mL of sterilized physiological saline solution (0.85%). It was centrifuged at 4 °C and 1000 rpm for 2 min, then the supernatant was allowed to settle and a dilution was prepared. A spreader was used to spread the dilution evenly on nitrogen-free medium (abbreviated as NFM) [15] and Pikovskaya’s (abbreviated as PKO, and the source of inorganic P was 5.0 g·L^−1^ Ca_3_(PO_4_)_2_) [16] agar plates. The culture dishes were incubated at 28 °C in an incubator for 7 days. Single colonies with transparent zones (phosphate-solubilizing zones) on the PKO agar plates and colonies with rapid growth on the NFM agar plates were selected and purified. The nitrogenase activity of the strains was determined using the acetylene reduction assay [17]. The strains were inoculated into PKO liquid medium and cultured for 12 days. The phosphorus content of the supernatant was determined using the molybdenum blue colorimetric method [18]. Finally, these strains, characterized by their N_2_-fixation and phosphate solubilization, were inoculated into King’s B liquid medium (Qingdao Haibo Biotechnology Co., Ltd., Qingdao, China) and cultivated at 28 °C with an agitation speed of 180 rpm for 12 days. A standard curve was plotted by adding IAA powder to prepare solutions of different concentrations. Afterwards, the characteristic of IAA production by the strains was quantitatively characterized using spectrophotometry [19]. 

The results concerning the bacterial isolation and growth-promoting characteristics are shown in Appendix A. Based on the nitrogen fixation, phosphate solubilization, and indole-3-acetic acid (IAA) production characteristics, four strains with excellent growth-promoting properties (HL3, HL6, HG12, HG24) were obtained from the rhizosphere of halophytes. The identification of the bacteria using molecular biology techniques proceeded as follows: the total DNA was extracted from bacterial cells using a commercial DNA extraction kit (TSINGKE TSP102-50, provided by Qingke Biotechnology Co., Ltd., located in Wuhan, China), and PCR amplification of the 16S rDNA was performed using the universal bacterial primers 1492 R and 27 F [20]. The PCR amplification conditions were as follows: initial denaturation at 98 °C for 2 min; 38 cycles of 98 °C for 10 s, 55 °C for 15 s, and 72 °C for 15 s; and final extension at 72 °C for 5 min. Finally, the PCR products were sent to Qingke Biotechnology (Wuhan) Co., Ltd., for nucleotide sequencing and to obtain the 16S rRNA gene sequences of the selected strains. The obtained gene sequences were submitted to the NCBI (National Center for Biotechnology Information) GenBank and subjected to BLAST sequence comparison analysis. 

### 2.2. Salt Tolerance of Selected Strains

First, 100 μL of bacterial suspension of every strain was added to a Luria–Bertani (LB) medium (Qingdao Haibo Biotechnology Co., Ltd., Qingdao, China) containing varying concentrations of NaCl (0%, 2%, 4%, 6%, 9%, 12%). The cultures were then incubated with shaking at 180 rpm at 28 °C for 24 h. Then, the *OD*_600_ value of the culture medium were measured using a spectrophotometer, with each treatment replicated three times. Due to the inherent biological variability, the tolerance results of the PGPR strains are expressed as relative values. These were calculated as the ratio of the *OD*_600_ under stress conditions to the *OD*_600_ under non-stress conditions.

### 2.3. Qualitative Determination of Plant Growth-Promoting Characteristics of Selected Strains

The selected strains were re-inoculated onto Ashby’s agar plates [21] (Shanghai Ruichu Biotechnology Co., Ltd., Shanghai, China) and PKO agar plates [16] to observe their N_2_ fixation and phosphate solubilization characteristics, respectively. Simultaneously, the selected strains’ ability to solubilize organic phosphorus was qualitatively assessed using Mongina agar plates (the source of organic P was 0.2 g·L^−1^ lecithin) [16].

The strains were re-inoculated into King’s B liquid medium [22] and cultured for 72 h for the qualitative measurement of the bacterial IAA production characteristics. The procedure is outlined briefly as follows: an equal volume of Salkowski’s reagent (1 mL of 0.5 mol·L^−1^ FeCl_3_ dissolved in 50 mL of distilled water, with 30 mL of concentrated sulfuric acid added, and a volume made up to 100 mL) was mixed with an equal volume of the bacterial suspension to observe the color changes. A negative control (no inoculation and no IAA added) and a positive control (no inoculation but with 100 μg·mL^−1^ IAA added) were set up. Darker colors indicated the more vital ability of the strain to secrete IAA and, vice versa, the weaker ability.

The potential for EPS production by the strains was assessed on a special medium [23] (yeast extract: 20 g; KH_2_PO_4_: 15 g; MgSO_4_: 0.2 g; MgSO_4_: 0.015 g; FeSO_4_: 0.015 g; CaCl_2_: 0.03 g; NaCl: 0.015 g; agar: 15 g; sucrose: 100 g; distilled water: 1000 mL). The strains were inoculated onto the medium and incubated at 28 °C for 7 days. The ability of the strains to produce EPS was determined based on the presence or absence of mucilage formation around the colonies. 

The qualitative testing of the ability of bacterial strains to produce siderophores followed these steps [24]: the strains are inoculated onto Chrome Azurol S (CAS, Qingdao Haibo Biotechnology Co., Ltd., Qingdao, China) agar plates and incubated at 28 °C for 7 days. A color change zone around the colonies indicated the ability to produce siderophores. Subsequently, the bacterial culture was centrifuged to obtain the supernatant, which was then thoroughly mixed with an equal volume of CAS assay solution for 1 h. After this incubation, the optical density at 630 nm (OD630) was measured. The capacity to produce siderophores was calculated using the following formula:$$Siderophore\;production\left(\%\right)=\frac{A_{r}-A_{s}}{A_{r}}\times 100\%,$$
where $A_{r}$ = the absorbance of reference (CAS solution and uninoculated broth) and $A_{s}$ = the absorbance of the sample (CAS solution and cell-free supernatant of sample).

### 2.4. Quantitative Determination of Plant Growth-Promoting Characteristics of Selected Strains

The abilities of the selected strains to fix nitrogen, solubilize inorganic phosphorus, and produce IAA were re-evaluated by employing the techniques outlined in Section 2.1. Simultaneously, these selected strains were inoculated into Mongina liquid culture medium. Adhering to the protocols for the solubilization of inorganic phosphorus by bacteria, their capacity to solubilize organic phosphorus was quantitatively assessed using the molybdenum blue colorimetric method [18]. The pH value of the PKO and Mongina liquid medium was also determined for the inoculated and uninoculated (control) bacterial strains.

The characteristics of the EPS production by the strains were quantitatively determined through the phenol–sulfuric acid method combined with a standard curve [8]. The test strains were inoculated into 50 mL of LB liquid medium and incubated at 28 °C for 24 h to activate. Subsequently, 1% of the culture was inoculated into 50 mL of LB liquid medium and incubated at 28 °C for 24 h to expand the culture. The culture was centrifuged at 9000 rpm and 4 °C for 10 min to collect the bacterial cells. The cells were washed twice with DF liquid medium without (NH_4_)_2_SO_4_ by centrifugation, then resuspend in 25 mL of ADF medium with different NaCl concentrations (0 and 100 mM) and incubated at 28 °C for 48 h. Centrifugation was performed again at 9000 rpm and 4 °C for 10 min to collect the cells, which were washed twice with Tris-HCl buffer (pH 7.6, 0.1 mol·L^−1^). The cells were resuspended in 600 μL of 0.1 mol·L^−1^ Tris-HCl buffer (pH = 8.5), and 30 μL of toluene was added and shaken vigorously for 30 s to lyse the cells, yielding a crude enzyme solution. Then, 100 μL of the crude enzyme solution was stored at 4 °C for the protein determination; the remainder of the crude enzyme solution was used immediately for the ACC deaminase activity determination [15,22].

Two sets of experiments were conducted to quantitatively determine the phosphate solubilization, IAA, ACC deaminase, and EPS production by the bacterial strains. One set used the original culture medium formulation, and the other used a culture medium supplemented with 100 mM NaCl to compare the plant growth-promoting characteristics of the bacterial strains under different salt-stress conditions.

### 2.5. The Effect of Selected Strains on the Growth of Alfalfa in Saline Soil

#### 2.5.1. Preparation of Bacterial Inocula

The test strains were inoculated into 250 mL Erlenmeyer flasks containing 250 mL of LB liquid medium and incubated at 28 °C with constant shaking at 180 rpm for 20 h. After incubation, the culture was centrifuged, and the supernatant was discarded. The bacterial cells were washed twice with sterile water. Finally, the bacterial suspension’s optical density (*OD*_600_) was adjusted to 0.6 for further use.

#### 2.5.2. Pot Experiment

Sterile water was used for the control and inoculation treatments, while various bacterial suspensions served as the inoculation treatments. At the beginning of spring 2023, a pot experiment was conducted in the greenhouse of the Institute of Soil and Water Conservation, Chinese Academy of Sciences, and Ministry of Water Resources, located in Minqin County, Gansu Province, China. The pots used for the experiment had dimensions of 24 cm (top diameter) × 19 cm (bottom diameter) × 24.5 cm (height). Prior to use, these pots were disinfected twice with alcohol and air-dried. Topsoil from the local field (previously planted with corn) was collected from the tillage layer, thoroughly mixed, air-dried, and sieved. This soil was then packed into sterilization bags and subjected to high-temperature steam sterilization to reduce the impact of native soil microorganisms on the colonization and function of the plant growth-promoting rhizobacteria (PGPR). After sterilization, the soil was placed into the prepared pots and moistened with local agricultural irrigation water to the appropriate moisture level before seeding. The properties of the test soil and the salt content of the irrigation water are presented in Table 1.

The seeds were disinfected and washed, then soaked in the corresponding PGPR bacterial suspensions, with each treatment being replicated three times. The sowing date was set for 1 April 2023. Two weeks after germination, the seedlings were thinned to leave one plant per pot; then, 5 mL of the corresponding bacterial agent was added to the rhizosphere of each seedling. The plants were harvested 45 days and 90 days after sowing.

#### 2.5.3. Plant Growth and Biomass

The plant height was measured from the soil surface to the top bud using a ruler, which was recorded as the plant height. The leaves were removed, and the soil and other stains were carefully wiped off with a paper towel before laying them out individually on A4 paper. They were then scanned into fixed images using a scanner. The total number of leaves and the total leaf area were obtained by either manually counting them or using Image J software (v1.8.0, Image-adjust-Threshold), thereby calculating the average leaf area. The above-ground parts and roots were collected, placed in an oven at 80 °C, and dried until a constant weight was achieved, and then the dry weight was measured.

#### 2.5.4. Proline Content

Sulfosalicylic acid was used to extract proline from the plant roots. Briefly, 0.05 g of fresh leaves were weighed and chopped up, and 3% sulfosalicylic acid was added to obtain the extract. Then, 2 mL of the extract was taken, to which 2 mL of glacial acetic acid and 2 mL of acid ninhydrin were added, and it was placed in a water bath at 100 °C to boil for 1 h. After the reaction mixture had cooled, it was extracted with toluene, and the absorbance was measured. The proline content was determined by reading from a standard curve [15].

#### 2.5.5. Membrane Stability Index

Fresh roots (100 mg) were divided into two groups and placed in 5 cm^3^ of double-distilled water. One group was left at 40 °C for 30 min, and its electrical conductivity (C1) was recorded using a conductivity meter. The second group was maintained at 121 °C in a pressure cooker for 30 min, and its electrical conductivity (C2) was recorded. The membrane stability index (MSI) was calculated using the following formula [25]:$$MSI=\left(1-\frac{C1}{C2}\right)\times 100\%$$

### 2.6. Statistical Analysis

A one-way analysis of variance (ANOVA) was employed to analyze the effects of the different salinity treatments on the growth-promoting characteristics of the PGPR strains and the impact of the different bacterial inoculations on plant growth under salt stress. For multiple comparisons between groups, the Tukey test was used. Data are reported as the mean value (n = 3) ± standard deviation.

## 3. Results

### 3.1. Identification and Salt-Drought Tolerance of Selected Strains

The strains HL3, HL6, HG12, and HG24 were preliminarily identified as *Priestia filamentosa*, *Bacillus atrophaeus*, *Bacillus subtilis* subsp. *stercoris*, and *Paenibacillus peoriae*, respectively. The sequences obtained were submitted to the GenBank database, receiving accession numbers OQ683811, OQ683815, OQ683816, and OQ683817, respectively.

All the strains were capable of tolerating 2% NaCl. When the culture medium contained 2% NaCl, the growth of HL3, HL6, and HG12 was lower than in the absence of NaCl, while the growth of HG24 was more significant than in the absence of NaCl. HL3, HL6, and HG12 could maintain more than 75% of their growth activity under 9% NaCl stress compared to unstressed conditions. Under 12% NaCl stress, HL3 and HL6 could still maintain 50% of their growth, while the growth capacity of HG12 dropped to only 22% of the unstressed condition. Although the growth capacity of strain HG24 exceeded that without NaCl addition when 2% NaCl was present, the strain ceased to proliferate when the NaCl concentration reached 4% (Figure 1).

### 3.2. Plant Growth-Promoting Traits of Selected Strains

#### 3.2.1. Qualitative Analysis

Qualitative analysis revealed that HL3, HG12, and HG24 could grow on nitrogen-free media, qualitatively verifying their N_2_ fixation ability (Figure 2a, Table 2), while HL6 did not exhibit significant nitrogen-fixing characteristics. Around HL3, HL6, HG12, and HG24, clear zones appeared on the inorganic phosphate plates, indicating that all four strains possessed the characteristic of dissolving inorganic phosphates. Compared to the other three strains, the phosphate solubilization zone around HL3 was not prominent (Figure 2b, Table 2). On the organic phosphate solubilization plates, clear zones appeared around HL3, HL6, HG12, and HG24; HG24 showed a more substantial phosphate solubilization potential (Figure 2c, Table 2). All the strains had the potential to secrete IAA, with HG12 showing a lighter color, followed by HL3, while HL6 and HG24 did not show a significant color contrast (Figure 2d, Table 2). As indicated by a viscous substance around their colonies, HL6, HG12, and HG24 could produce EPS (Figure 3a, Table 2). In contrast, no such substance was observed around HL3, suggesting it lacked the potential to secrete large amounts of extracellular polysaccharides. On the CAS (Chrome Azurol S) medium, HL6 and HG24 exhibited distinct yellow halos (Figure 3b, Table 2), indicating their ability to produce siderophores, while HL3 and HG12 did not display similar characteristics.

#### 3.2.2. Quantitative Analysis

Quantitative analysis indicated that the nitrogenase activity of HL3, HG12, and HG24 ranged between 148.03 and 266.45 nmol C_2_H_4_·h^−1^, with strain HG24 exhibiting the highest nitrogenase activity (Figure 4f).

All four PGPR strains could solubilize inorganic phosphate. Under no salt addition, the phosphate solubilization ranged from 39.73 to 130.73 μg·mL^−1^, with strain HL3 having the highest inorganic phosphate solubilization capacity at 130.73 μg·mL^−1^. Under 100 mM NaCl salt stress, the phosphate solubilization ranged from 40.53 to 133.64 μg·mL^−1^, with strain HL3 showing the highest inorganic phosphate solubilization capacity. Except for HG24, compared to the control group (no salt addition), 100 mM NaCl salt stress significantly enhanced the inorganic phosphate solubilization capability of the strains (Figure 4a).

All four strains possessed the ability to solubilize organic phosphates. Under conditions without salt addition, the solubilization ranged from 1.61 to 3.47 μg·mL^−1^, with the strain HL6 showing the highest organic phosphate solubilization capacity of 3.47 μg·mL^−1^. Under 100 mM NaCl salt stress, the phosphate solubilization ranged from 1.68 to 3.49 μg·mL^−1^, with strain HL6 again showing the highest inorganic phosphate solubilization capacity of 3.49 μg·mL^−1^. Compared to the control group (no salt addition), 100 mM NaCl salt stress significantly enhanced the organic phosphate solubilization ability of strain HL3 but significantly inhibited that of HG24; the organic phosphate solubilization abilities of HL6 and HG12 showed no significant difference under the two salt treatments (Figure 4b).

Under conditions without salt addition, all four strains could produce a certain amount of IAA, ranging from 21.15 to 35.15 μg·mL^−1^, with HG24 secreting the highest amount of IAA. Under 100 mM NaCl salt stress, the amount of IAA produced by each strain ranged from 7.83 to 20.64 μg·mL^−1^, with strain HL6 showing the highest IAA secretion. Compared to the control, salt stress significantly inhibited the IAA secretion characteristics of all the PGPR strains (Figure 4c).

Under conditions without salt addition, all four strains exhibited a specific capacity for ACC deaminase production, with HL6 showing the highest activity of 40.88 μmol α-ketobutyrate·mg^−1^·h^−1^. Moreover, 100 mM NaCl stress significantly reduced the ACC deaminase activity of strains HL3 and HL6 but enhanced the ACC deaminase activity of strain HG12. HG24 displayed lower ACC deaminase activity under both salt treatment conditions, with no significant difference between the two groups (Figure 4d).

Under conditions without salt addition, HL6, HG12, and HG24 could produce EPS. Compared to the control group (no salt addition), 100 mM NaCl stress significantly promoted EPS production in strain HL6 but inhibited EPS secretion in strain HG24. HG12 displayed lower EPS production capacity under both salt treatment conditions, with no significant difference between the two groups (Figure 4e). Only strains HL6 and HG24 could produce a certain amount of siderophores, with HG24 producing more than HL6 (Figure 4g).

### 3.3. Effects of Inoculation with Selected Strains on Alfalfa Growth

During the seedling stage, compared to the control, PGPR inoculation did not significantly affect the height, total leaf number, total leaf area, average leaf area, shoot biomass, root biomass, total biomass, or root-to-shoot ratio of the alfalfa seedlings. However, it significantly increased the membrane stability (HG12) and proline content (HL3, HL6, HG12) (Figure 5).

During the flowering stage, the impact of PGPR was negative, neutral, or positive. Compared to the control, inoculation with HL6, HG12, and HG24 significantly increased the plant height (Figure 5a), and inoculation with HL6 and HG12 significantly increased the total number of leaves (Figure 5b). However, inoculation with all the PGPR strains did not significantly affect the total and average leaf area (Figure 5c,d). Interestingly, compared to the control, inoculation with HL3, HL6, and HG12 significantly increased, but inoculation with HG24 decreased, the dry weight of the alfalfa shoots, roots, and total dry weight (Figure 5e–g); inoculation with HL3 and HL6 significantly increased the root-to-shoot ratio, while HG12 inoculation had no significant effect on the root-to-shoot ratio, but HG24 inoculation reduced it (Figure 5h). None of the inoculation treatments significantly affected the membrane lipid stability (Figure 5i). All the inoculation treatments significantly reduced the proline content of the alfalfa tissues (Figure 5j).

There were significant differences in the plant height, leaf number, leaf area, biomass, and root-to-shoot ratio between alfalfa’s seedling and flowering stages. However, the two growth stages had no statistical difference in terms of the membrane lipid stability index. The changes in the proline content varied: in the control and HG24 inoculated groups, the proline content inside the tissues during the flowering stage was higher than in the seedling stage, while in the HL3, HL6, and HG12 inoculation treatments, there was no statistical difference in the proline content between the two stages (Figure 5). From both the physiological regulation and growth perspectives, strains HL3, HL6, and HG12 were viable bioagent resources that could be applied to the saline–alkali soils of Northwest China to enhance plant production.

## 4. Discussion

### 4.1. Inconsistencies in Strain Characteristic Analysis between Qualitative Testing and Quantitative Testing

In the qualitative and quantitative analyses, the growth-promoting characteristics of the PGPR strains, such as nitrogen fixation, siderophore production, extracellular polysaccharide (EPS) production, and indole-3-acetic acid (IAA) production abilities, showed consistent results. However, differences were observed in the phosphate solubilization characteristics (Figure 2 and Figure 4). Compared to the other three strains, HL3 did not exhibit a clear phosphate solubilization zone on the agar plates, yet it demonstrated higher phosphate solubility in the culture medium (Figure 4a). On the organic phosphate solubilization plates, HL6’s solubilization zone was not as straightforward as HG24’s, but its supernatant had the highest effective phosphate concentration among the four strains (Figure 4b). This indicates that characterizing bacteria through qualitative tests on plates has limitations [26]. Phosphate-solubilizing bacteria exhibit different adaptabilities to plate cultivation and liquid culture; some strains reproduce faster and have more substantial phosphate solubilization capabilities in liquid culture [27]. The inconsistency between the size of the solubilization zone and the amount of solubilized phosphate may also be due to the complexity of the strains’ phosphate solubilization mechanisms [28]. The appearance of a solubilization zone primarily reflects the production of organic acids by phosphate-solubilizing microbes [29]. Under acidic conditions, components such as Ca_3_(PO_4_)_2_ and CaCO_3_ in the medium dissolve, leading to solubilization zones or an increase in the effective phosphate content in the liquid medium [29,30,31]. Studies have shown that the activity of phosphate-solubilizing bacteria is significantly negatively correlated with the pH value of the culture medium [32,33]. In this study, compared to the control group, the pH of the inorganic phosphate medium significantly decreased with PGPR inoculation (Figure 6a). However, there was no significant correlation between the phosphate solubilization ability and the pH of the culture medium (Figure 6c), indicating that other mechanisms, such as enzymatic action and protein mediation, play a role in phosphate solubilization [31]. Bacteria that solubilize organic phosphate do so by secreting extracellular phosphatases and other non-organic acid substances to decompose organic phospholipids [34,35]; hence, their metabolic products do not significantly affect the pH of the culture medium (Figure 6d). However, the pH of most strain cultures also decreases (Figure 6b), suggesting that the acid-producing characteristics of the strains may also contribute to phosphate solubilization. Organic acids reduce the soil pH and make the environment more acidic, promoting hydrolysis and the release of organic phosphorus. In addition, organic acids can form complexes with organophosphorus, changing their chemical properties and making them more susceptible to microbial degradation [36].

The plate assay method is a simple and rapid preliminary screening technique that helps identify PGPR strains with potential growth-promoting activities [37]. However, the growth-promoting potential of PGPR needs further evaluation through liquid culture medium measurements. This method is generally more precise than plate measurements and can more accurately assess the potential of PGPR strains to enhance plant growth [38]. For instance, the potential for IAA production by HL6, HG12, and HG24 was not distinctively apparent on porcelain plates (Figure 2d), coupled with the biological characteristics of the PGPR, where bacterial spread could inundate or cover the plate, making it difficult to discern the required growth-promoting traits visually (Figure 3b). Therefore, the growth-promoting characteristics can only be determined through quantitative testing of the fermentation broth.

### 4.2. Significant Inhibition of IAA Production Characteristics in Strains under Salt Stress Compared to Other Growth-Promoting Traits

Existing studies rarely report the changes in the growth-promoting traits of PGPR under salt addition [12,39,40], instead inferring their growth-promoting characteristics based on the strains’ salt tolerance. Research has also shown that if a strain strongly resists adverse conditions, its conferred growth-promoting traits are either unaffected or less affected [41]. However, the situation is not so absolute. Although strain HG24 could not tolerate 4% NaCl in this study, the other strains’ highest tolerance range could extend from 9% to 12% (Figure 1). The IAA secretion characteristic of all the strains was negatively affected by 100 mM NaCl stress, but the phosphate solubilization, ACC deaminase production, and EPS functions of some strains were enhanced by salt addition (Figure 4). This may be because (1) under salt-stress conditions, PGPR may prioritize adjusting their metabolic pathways to adapt to the environment, affecting the synthesis and secretion of IAA. The biochemical pathways involved in nitrogen fixation and phosphate solubilization may respond differently to salt stress; they may have higher salt tolerance mechanisms or respond more slowly to salt stress [42]. (2) Salt may directly or indirectly affect the stability and bioavailability of IAA. In contrast, the nitrogen fixation and phosphate solubilization activities are crucial for plant growth, especially under nutrient-deficient conditions; thus, these functions may be preserved or regulated through other mechanisms to adapt to salt stress. The results presented in this paper also suggest that the ability of the strains to secrete IAA can serve as a preferred indicator for the preliminary screening of salt-tolerant PGPR, which would significantly improve the efficiency of selecting salt-tolerant PGPR [43].

### 4.3. The Varied Role of Selected Strains at Different Growth Stages of Alfalfa

Plant functional traits such as the plant height, leaf area, and biomass are readily quantifiable indicators that effectively reflect plant survival strategies [44,45]. The Membrane Stability Index (MSI) measures the stability of plant or other organismal cell membrane lipids under stress, directly affecting plant growth and adaptability. A higher MSI value typically indicates excellent cell membrane stability against stress, while a lower MSI value suggests the cell membrane is more susceptible to damage [46]. Proline plays a crucial role in plant stress resistance. The proline content significantly increases when plants face adverse environmental stresses like drought, salt, and cold. It helps plants maintain cell osmoregulation, protect cell membrane integrity, regulate redox balance, and reduce free radical damage, enhancing plants’ stress resistance [47].

In this study, the potting soil was mildly saline–alkaline, and the irrigation water was slightly brackish, indicating that salt stress was present throughout the entire growth period of the alfalfa plants. PGPR inoculation did not affect the growth indicators of the alfalfa seedlings but significantly impacted physiological indicators such as the membrane lipid stability and proline content, especially for the strain HL6 (Figure 5). Compared to the seedling stage, the effects of PGPR inoculation were relatively inconsistent during the flowering stage. For example, most treatments significantly altered growth indicators such as the plant height, leaf number, and biomass but did not significantly affect physiological indicators like the MSI; the proline content of HL3, HL6, and HG12 was significantly increased compared to the control group, and their corresponding biomass also increased. This may be because the improvement of plant development requires more proline to support metabolic activities and growth demands [48,49]. The physiological changes in alfalfa growth mediated by PGPR at different growth stages suggest that PGPR plays different roles or adopts different strategies at various plant growth stages, affecting host plant growth.

During the seedling stage, plants have weaker resistance. The root system is the first part to experience salt stress, which limits the roots’ ability to absorb water [6,7]. Studies have shown [8,42] that PGPR can mitigate the impact of salt stress by synthesizing extracellular polymeric substances (EPS) that chelate Na+, reducing root ion toxicity and osmotic stress. Additionally, PGPR can synthesize ACC deaminase to degrade the ethylene precursor ACC [7,37,50,51]. HL6 and HG12 exhibited strong EPS and ACC deaminase secretion characteristics in this study. It is speculated that the alleviation of salt-stress damage may be related to the secreted EPS and ACC deaminase.

Furthermore, PGPR can also alleviate plant damage by inducing the synthesis of hormones like abscisic acid (ABA) or ethylene in the host plant, which closes the leaf stomata to reduce water transpiration [52,53]. When subjected to salt stress, salt-sensitive plants may reduce root growth to avoid extensive contact with salt, mitigating salt damage [54,55]. During the seedling stage, the root-to-shoot ratio of the PGPR inoculated group was higher than that of the control group but not significantly, suggesting that the plants were under salt stress and that PGPR inoculation could mitigate salt stress. However, this mitigating effect is not solely reflected in the root-to-shoot ratio; it could also be related to alleviating oxidative and osmotic stress (Figure 5h–j). In summary, during the seedling stage, the role of PGPR is to help plants cope with salt stress; the mechanism adopted involves inducing physiological changes to achieve good adaptation to salinity.

As the resistance of alfalfa to adverse conditions strengthens, PGPR gradually mitigates the constraints of salinity. Inoculation with HL3 and HL6 significantly increased the root-to-shoot ratio, whereas HG12 inoculation had no significant effect on the root-to-shoot ratio, but HG24 inoculation reduced it (Figure 5h). Changes in the root-to-shoot ratio suggest that the root system grows adequately to absorb soil nutrients, thereby accumulating biomass, an important indicator of increased plant resistance [56,57]. During this period (flowering stage), plants are less or not stressed; hence, there was no significant difference in the MSI between the control and treatment groups, suggesting that the mechanism of PGPR influence might be to dissolve soil nutrients and produce growth hormones to promote host plant growth [7] rather than physiological regulation to increase adaptability. This evidence includes (1) no significant difference in the MSI between different treatments (Figure 5i); (2) a significant reduction in the proline content compared to the control group; and (3) the treatment groups with the most biomass accumulation (HL3, HL6, HG12) had the lowest proline content.

Our research results indicate that the PGPR strains can enhance the salt tolerance of alfalfa seedlings and promote biomass accumulation. Considering salt tolerance, growth-promoting traits, and growth-promoting effects comprehensively, HL3, HL6, and HG12 have great potential as microbial inoculants in arid and semi-arid regions. Further research should explore the possibility of using these bacteria and their consortia as microbial inoculants for crops in saline–alkaline soils, including the colonization rate of PGPR changes in soil and the microbial community characteristics mediated by PGPR.

## 5. Conclusions

Combining qualitative and quantitative analyses and plant growth measurements defined the multifaceted growth-promoting functions of PGPR. Compared to the control group (no additional NaCl added), under 100 mM NaCl, the IAA production of the selected strains significantly decreased. In contrast, the changes in phosphate solubilization, ACC deaminase activity, and extracellular polysaccharide (EPS) production varied inconsistently among the strains (either increased or decreased). The PGPR strains improved the alfalfa seedlings’ morphological and physiological characteristics, with the effects of PGPR inoculation on alfalfa varying between the seedling and initial flowering stages. During the seedling stage, the role of PGPR is to regulate metabolism to help plants cope with salt stress; in the flowering stage, the impact of PGPR is primarily manifested in increased biomass accumulation. This study contributes to guiding the screening and application of PGPR. However, further evidence is required to confirm how PGPR mitigates salinity stress.

## Figures and Tables

**Figure 1 microorganisms-12-00616-f001:**
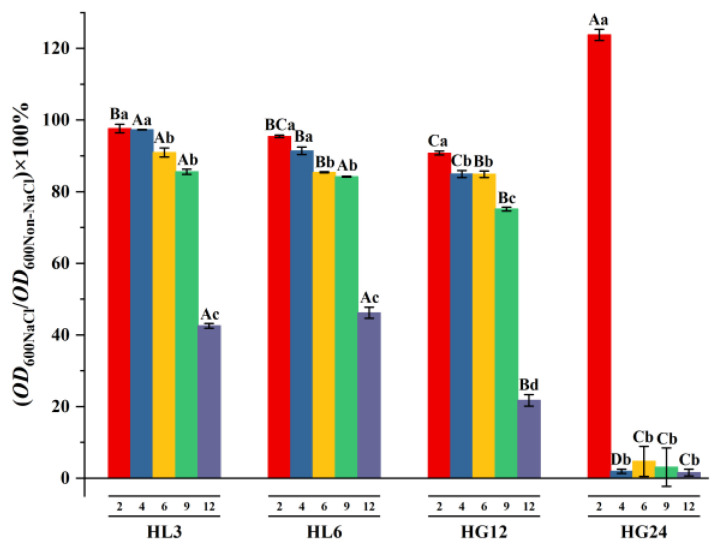
Relative growth of the selected strains at different concentrations of NaCl solution. Here, 2, 4, 6, 9, and 12 represent 2%, 4%, 6%, 9%, and 12% NaCl solutions, respectively. Different uppercase letters indicate differences in salt tolerance among different PGPR strains under the same salt treatment; different lowercase letters indicate differences in salt tolerance for the same PGPR strain under different salt treatments.

**Figure 2 microorganisms-12-00616-f002:**
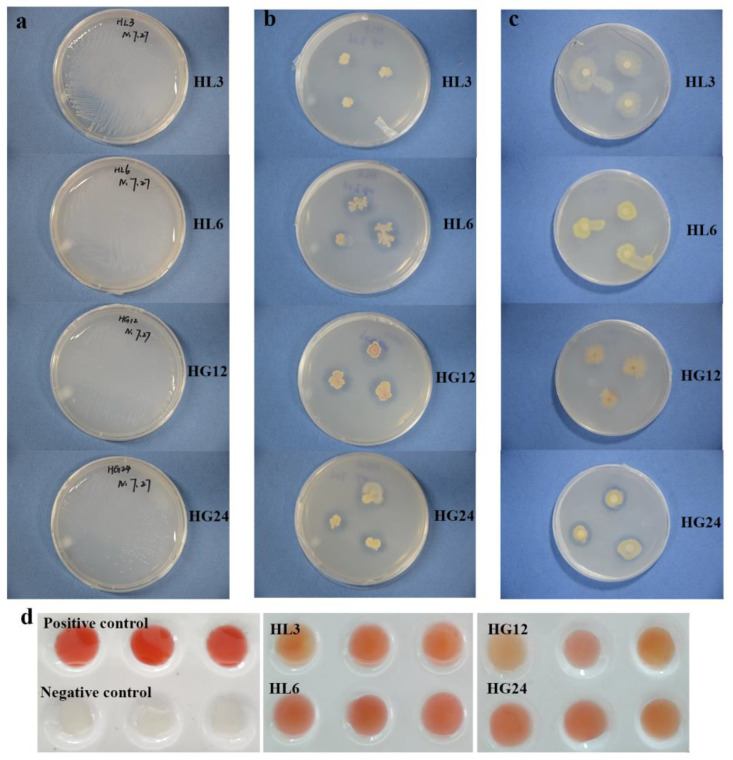
Qualitative determination of N_2_ fixation (**a**), inorganic phosphorus solubilization (**b**), organic phosphorus solubilization (**c**), and IAA secretion (**d**) by the selected strains.

**Figure 3 microorganisms-12-00616-f003:**
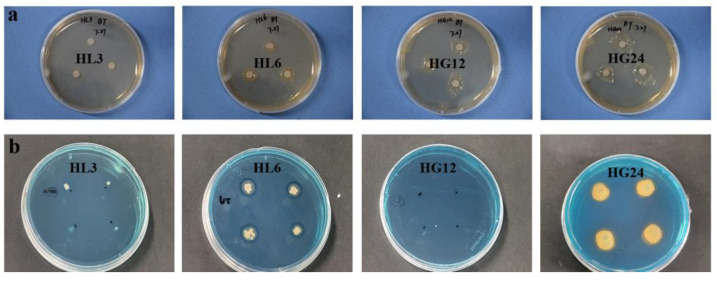
Qualitative determination of EPS (**a**) and siderophores (**b**) produced by the selected strains.

**Figure 4 microorganisms-12-00616-f004:**
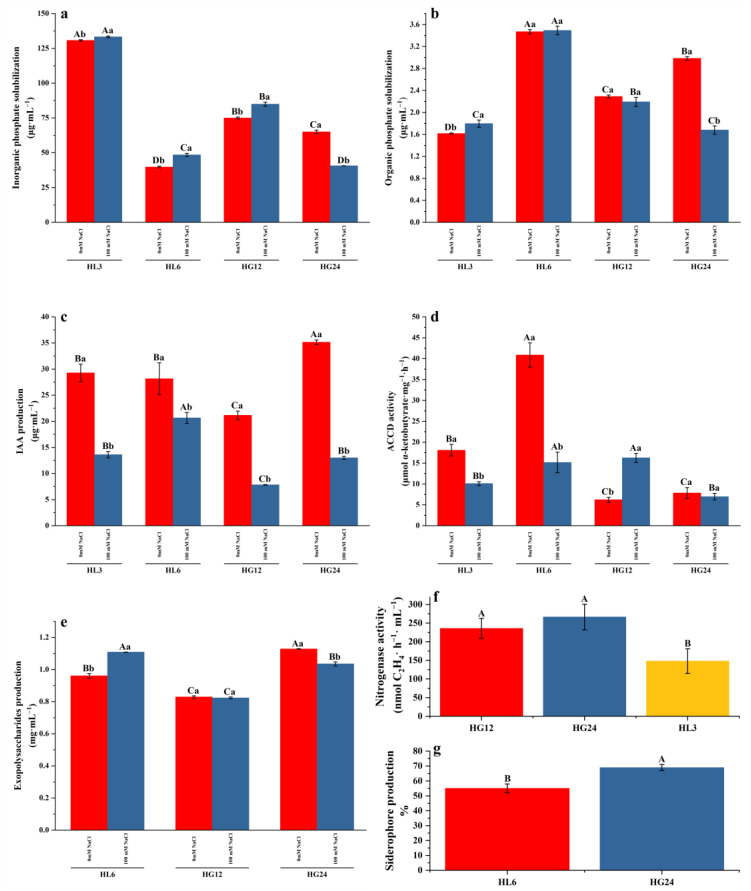
Inorganic phosphorus solubilization capacity (**a**), organic phosphorus solubilization capacity (**b**), IAA production (**c**), ACC deaminase activity (**d**), and EPS production (**e**) of the selected strains with 0 and 100 mM NaCl. Nitrogenase activity (**f**) and siderophore production (**g**) of the selected strains without NaCl. Uppercase letters indicate differences in growth-promoting characteristics among different strains; lowercase letters indicate differences in growth-promoting characteristics of each strain under different salt treatments.

**Figure 5 microorganisms-12-00616-f005:**
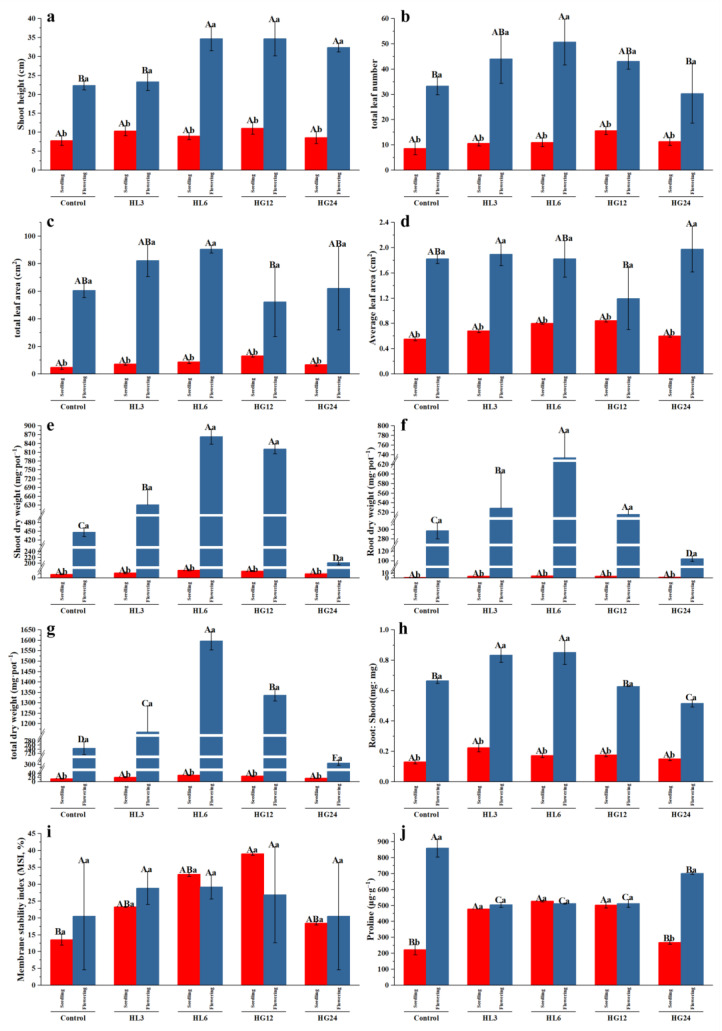
Effects of inoculation with the selected strains on the (**a**) shoot height, (**b**) total leaf number, (**c**) total leaf area, (**d**) average leaf area, (**e**) shoot dry weight, (**f**) root dry weight, (**g**) total dry weight, (**h**) root: shoot, (**i**) MSI, and (**j**) proline. Uppercase letters indicate differences in growth-promoting characteristics among different strains; lowercase letters indicate differences in growth-promoting characteristics of each strain under different salt treatments.

**Figure 6 microorganisms-12-00616-f006:**
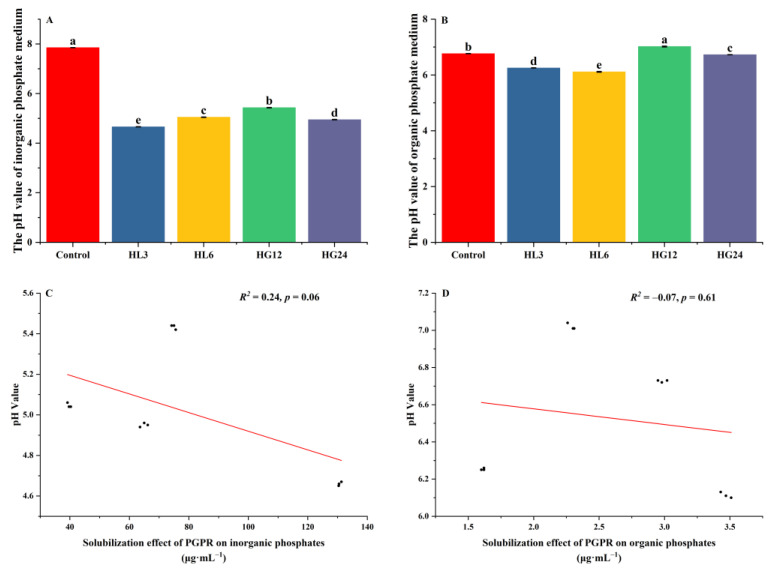
The phosphorus solubilization activity of the selected strains and its relationship with the pH value of the culture medium. (**a**) The pH value of the bacterial inorganic phosphorus culture medium; (**b**) the pH value of the bacterial organic phosphorus culture medium; (**c**) the correlation between the inorganic phosphorus solubilization activity and pH value of the bacterial culture medium; and (**d**) the correlation between the organic phosphorus solubilization activity and pH value of the bacterial culture medium. Lowercase letters indicate differences in the pH value of the bacterial suspensions of the different strains.

**Table 1 microorganisms-12-00616-t001:** The properties of the soil and irrigation water.

Soil Properties (n = 4)	Value	Irrigation Water Properties	Value
Soil texture	Silt loam (clay: 8.52%; silt: 54.53%; sand: 36.94%)	EC (dS·m^−1^, n = 3)	2.09 ± 0.01
EC_soil:water=1:5_(dS·m^−1^)	0.835 ± 0.038	TDS (%)	0.10%
pH	8.47 ± 0.01	pH (n = 3)	7.82 ± 0.09
Salt content (%)	2.73 ± 0.12	Na^+^ (mg·L^−1^, n = 3)	262.19 ± 1.22
Organic matter (g·kg^−1^)	7.26 ± 0.30	Ca^2+^ (mg·L^−1^, n = 3)	108.01 ± 0.85
Total nitrogen (g·kg^−1^)	0.67 ± 0.03	Ma^2+^ (mg·L^−1^, n = 3)	53.48 ± 0.16
Available nitrogen (mg·kg^−1^)	57.06 ± 3.79	K^+^ (mg·L^−1^, n = 3)	2.57 ± 0.11
Total phosphorus (g·kg^−1^)	0.62 ± 0.03	Cl^−^ (mg·L^−1^, n = 3)	272.48 ± 28.88
Available phosphorus (mg·kg^−1^)	13.97 ± 0.51	NO_3_^−^ (mg·L^−1^, n = 3)	5.99 ± 0.44
Available potassium (mg·kg^−1^)	99.09 ± 18.71	SO_4_^2−^ (mg·L^−1^, n = 3)	369.59 ± 34.52

**Table 2 microorganisms-12-00616-t002:** Qualitative analysis of the growth-promoting characteristics of the selected bacterial strains.

Plant Growth-Promoting Traits	HL3	HL6	HG12	HG24
N_2_ fixation	+	-	+	+
Inorganic phosphorus solubilization	+	+	+	+
Organic phosphorus solubilization	+	+	+	+
IAA production	+	+	+	+
EPS production	-	+	+	+
Siderophore	-	+	-	+

## Data Availability

All data generated or analyzed during this study are included.

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
