# Peer review of "Comparative Analysis of Plant Growth-Promoting Rhizobacteria’s Effects on Alfalfa Growth at the Seedling and Flowering Stages under Salt Stress"

_microorganisms, 2024, doi:10.3390/microorganisms12030616_

Round 1

Reviewer 1 Report

Comments and Suggestions for Authors

Comments to the Authors

The paper entitled “Comparative Analysis of PGPR’s Effects on Alfalfa Growth at Seedling and Flowering Stages Under Salt Stress” describes the potential of four plant growth-promoting (PGP) strains to ameliorate salt stress in Medicago sativa L.  The paper needs considerable improvement before acceptance for publication, and these improvements are particularly needed in the Introduction section, Material and Methods, and generally in terms of style and presentation. The authors should put an effort into rewriting, adding some clarifications, and additionally supporting and emphasize the novelty of the research conducted.

Here are listed major points and other minor suggestions:

1.      Abstract:

The English language has to be revised and any abbreviation should be defined at first mention (ACCD).

Line 21: IAA production was reduced in which strain? It should be clear.

2.      Introduction

The Introduction section should be completely revised since it is too broad. The general part of the introduction is too long, it does not focus on what is already known about the interaction of alfalfa with PGP, in saline environment or in general. From the Introduction part, the current state of research is not evident, it is difficult to identify the gaps, and which of them are progressing with the presented research.

Line 62-64: Are these results published or not? Are they included in the paper? If not, they are not sufficiently relevant for the Introduction.

3.      Materials and Methods

The origin of the strains as well as their isolation, selection, and identification should be explained in more detail. Preliminary screening mentioned in line 104 should be explained or the authors should refer to the literature describing the screening process.

Line 110-111: Which primers? Provide briefly necessary data about PCR reaction.

Line 114-115: Biochemical analyses changed or influenced on conclusion about strains’ identity obtained by molecular identification? If not, there is no need to use it. If they revealed the approximate identity of the strains, then maybe you could use more specific primers afterward, not universal ones.

Line 117: the amount of suspension is not important if we do not have the density

Title 2.3. “Plant Growth Promoting Traits of PGPR”, should be changed to “Plant Growth Promoting Traits of selected strains”

For each media used in the Material and Methods manufacturer data should be added.

Line 142-144: revise this part by describing briefly the method and choose one reference for siderophores.

Line 146: What do you mean by “activated bacterial suspensions” (should it be culture?) as well as by “corresponding liquid”?

Lines 147-149: not clear enough, add an explanation for each of the conducted quantifications.

Lines 150-162: use passive, not imperative in describing the method. Same for 2.5.1., 2.5.2., 2.5.3., and 2.5.4. paragraphs

Table 1 is not clear enough and should be completely revised. It is not clear which characteristics refer to soil and which are the irrigation water traits. Table title should be somewhat like “The properties of soil and irrigation water”, and title headings should be organized in a way that makes results more readable and understandable.

4.      Results

Results on identification: explain the role of traditional identification methods in final identification or exclude that part.

Figure 2: Add explanation for “2 4 6 9 12”

Section 3.2.1. It would be better to add a table showing all the tested PGP traits for used strains.

Line 275: exclude “fermentation experiment”

Line 289-298: The terminology used for traits related to P availability is confusing, making that part difficult to follow. Bacteria can mineralize organic P compounds, or solubilize inorganic phosphates. It is unclear what was analyzed and how. What do you mean by “solubilize organic phosphates”? Also, in Fig 5 a) and b) should be precisely explained.

5.      Discussion:

Line 365: It is not clear in which culture medium.

Line 388-389: It is well known that acid production is of importance in solubilization of inorganic phosphates, but its role in organic P mineralization should be explained and supported.

Line 423: You determined IAA production, and here you discuss about stability and bioavailability of IAA. Explain.

Line 426 (3) part): It seems that what was said under 3) is not related to exogenous IAA, but rather explains plant mechanisms, which is a completely other topic.

                                                                                                                                                          Sincerely,                                                                                                                             Reviewer

Comments on the Quality of English Language

Moderate editing is needed.

Reviewer 2 Report

Comments and Suggestions for Authors

The manuscript titled "Comparative Analysis of PGPR's Effects on Alfalfa Growth under Salt Stress" provides a comprehensive investigation into the impact of plant growth-promoting rhizobacteria (PGPR) on alfalfa growth in saline-alkaline soils. The study delves into the economic and ecological benefits of cultivating alfalfa in challenging environments, shedding light on the potential of PGPR to enhance plant growth and salt tolerance.

    The research conducted by the authors explores the differences in growth and physiological characteristics of alfalfa at various growth stages with different PGPR inoculations. Through a series of experiments, significant findings were observed, including variations in plant height, leaf number, leaf area, biomass, and root-to-shoot ratio between alfalfa seedling and flowering stages. Interestingly, the membrane lipid stability index did not show statistical differences between the two growth stages, indicating a consistent response to PGPR treatments.

    Moreover, the study highlights the role of proline content in alfalfa tissues under different PGPR treatments. The changes in proline content varied across inoculation groups, with some treatments showing higher proline content during the flowering stage compared to the seedling stage. This nuanced analysis provides valuable insights into the physiological responses of alfalfa to PGPR inoculations at different growth stages.

To improve the manuscript, consider the following suggestions:
 *Include a more detailed comparison with existing studies to highlight the novelty and contributions of your work. Discuss how your findings align or differ from previous research and the implications of these differences.
*Elaborate on the mechanisms through which PGPR strains exert their beneficial effects on alfalfa under salt stress. This could involve further exploration of the molecular interactions between PGPR and alfalfa, potentially through gene expression analysis or other molecular biology techniques.
*Expand on the environmental and economic benefits of using PGPR in agriculture, particularly in regions affected by salinization. This could involve a discussion on sustainability, reduction in chemical fertilizer usage, and potential for increased yields and farmer income.
* Discuss the potential application of these findings beyond alfalfa, including other crops that might benefit from PGPR inoculation, especially in saline-alkaline soils. This could include considerations for crop rotation practices or integrated pest management systems.
   *Consider adding a section on the long-term effects of PGPR inoculation on soil health and plant resilience. This could involve follow-up studies or models predicting long-term benefits or challenges.
*Clearly outline the limitations of the current study and propose future research directions. This might include exploring the effectiveness of PGPR in various soil types, climates, or with different levels of salinity stress.
*Provide guidelines or recommendations for the practical application of PGPR inoculation in agricultural practices. This could involve instructions for farmers on how to apply PGPR, expected costs, and management practices to maximize benefits.

In conclusion, this manuscript contributes significantly to the understanding of how PGPR can positively influence alfalfa growth under salt stress conditions. By elucidating the mechanisms underlying PGPR-mediated improvements in plant growth and salt tolerance, the study paves the way for developing sustainable agricultural practices that enhance crop productivity in saline-alkaline soils.

Comments on the Quality of English Language

The manuscript is written in English that is generally clear and understandable. Careful proofreading to correct any grammatical errors or punctuation mistakes is essential. This includes checking for subject-verb agreement, correct use of articles, and proper punctuation within and between sentences.

A thorough review by a native English speaker or a professional academic editor specializing in your field can provide valuable insights into improving the language and presentation of your manuscript.

Reviewer 3 Report

Comments and Suggestions for Authors

The present research evaluates the impact of 4 PGPR on the resistance of Alfalfa seedling and plant growth to salinity stress.

The introduction es well written with good references and present clearly the state of the art and the objectives of the reserach.

The material and method is complete

the results are well presented with a good description of the tables and figures.

The discussion is complete with good references.

Conclusions and abstract clear and coherents with the results obtained.

Minor corrections could be found in the attached document

Round 2

Reviewer 1 Report

Comments and Suggestions for Authors

The authors put a significant effort to improve the manuscript, so technically I have only several minor points. However, there is one very important issue that should be solved by editor.

In the first revision, the authors were asked to add necessary missing details about the isolation and selection of strains. Still, according to the reply, it is not possible as these results have been submitted to another journal, and the evaluation is in progress. This leads to a problematic situation, as either the reference to the published paper has to be made, or a description of this part should be added. Adding a short description, as they have decided to do, will ultimately lead to duplication of the published results, and it is always required to disclose that submitted results are not under evaluation by any other journal. In such situations, the widely accepted practice is to wait for the first paper (representing the first stage of the same study) to be published, and then refer to the paper in the second paper. Otherwise, I see no solution to this problem.

Minor corrections:

Line 22: “Inconsistent” change with “inconsistent”

Line 58-59: “Few studies compare the growth and physiological characteristics at different growth stages following PGPR inoculation.” Add references at the end of this sentence. Or add “few of these” if you rely on the previous sentence.

Line 64: 1st aim: it is better to say “of the selected PGPR strains”

Line 71: reference no.13 is not needed.

Line 74: add details about the commercial kit

Line 93-96: this part should be simplified, and the use of media for this purpose should be supported by the literature.

Line 97-100: It was not possible to find more information about Mongina medium. Add reference, as requested in a previous comment, as well as a source of phosphorus in media. In the response to the reviewer, I saw there is an explanation about the source of organic P (phospholipids), and it should be added to the manuscript.

Line 113-114: The confirmation test is usually needed afterward (precipitate formation in ethanol)
